# Blockchain and Firm Total Factor Productivity: Evidence from China

Qilong Cao [1,*,†], Jinglei Li [2,†], Hongru Zhang [2], Yue Liu [1] and Xun Luo [1]

1. Wu Jinglian School of Economics, Changzhou University, Changzhou 213000, China
2. School of Business, Changzhou University, Changzhou 213000, China
* Correspondence: cql1086@126.com; Tel.: +86-188-6124-3110
† These authors contributed equally to this work.

**Abstract:** This paper creatively constructs blockchain development indicators using geographical characteristics to investigate the influence of blockchain development on the total factor productivity of listed companies. Our findings reveal that local blockchain development can significantly promote the improvement of the firms' total factor productivity. To alleviate endogeneity, this paper combines exogenous policy and geographic distance to construct instrumental variables. Moreover, the positive influence is more pronounced in non-SOEs, non-excess capacity industries, and samples with high initial productivity. After the robustness test, the results are still valid. The aforementioned results provide practical implications for Chinese listed companies to lay out digital business.

**Keywords:** digital economy; blockchain technology; total factor productivity; Chinese listed companies

## 1. Introduction

With the in-depth advancement of the digital economy, digital technology has received worldwide attention [1,2]. Digital technology is the foundation of the digital economy, among which blockchain, big data, cloud computing, and artificial intelligence technologies are concerned to technology upgrades [3,4].

In the established literature, studies have focused on the influence of artificial intelligence [5,6] and 5G technology [7] on corporate performance, while few studies have specifically focused on blockchain technology's effect on firm production. In the paper, we counted the number of companies involved in blockchain technology in the city and constructed an indicator to measure the development of blockchain technology. The findings indicate that the development of blockchain technology has a positive impact on the total factor productivity of enterprises (hereafter referred to as TFP), and the positive influence is more pronounced in non-SOEs, non-excess capacity industries, and samples with high initial productivity.

Blockchain is a decentralized digital ledger with superior advantages in improving transaction efficiency and protecting information security and transparency; it is considered by governments as a disruptive technology [8,9]. Since 2016, the Chinese government has released a series of policies to promote the development of blockchain technology to accelerate the application of blockchain technology in Chinese enterprises. Figure 1 shows the number of newly registered blockchain companies per month in 2016–2019. Listed companies have blockchain technology with two modes of participation. The first is independent research and development. For example, dotcom firms such as Lenovo and Alibaba have established their own blockchain platforms. The second is shareholdings, where companies do not directly participate in the research and development of blockchain technology, but participate in the competition by investing in blockchain companies.

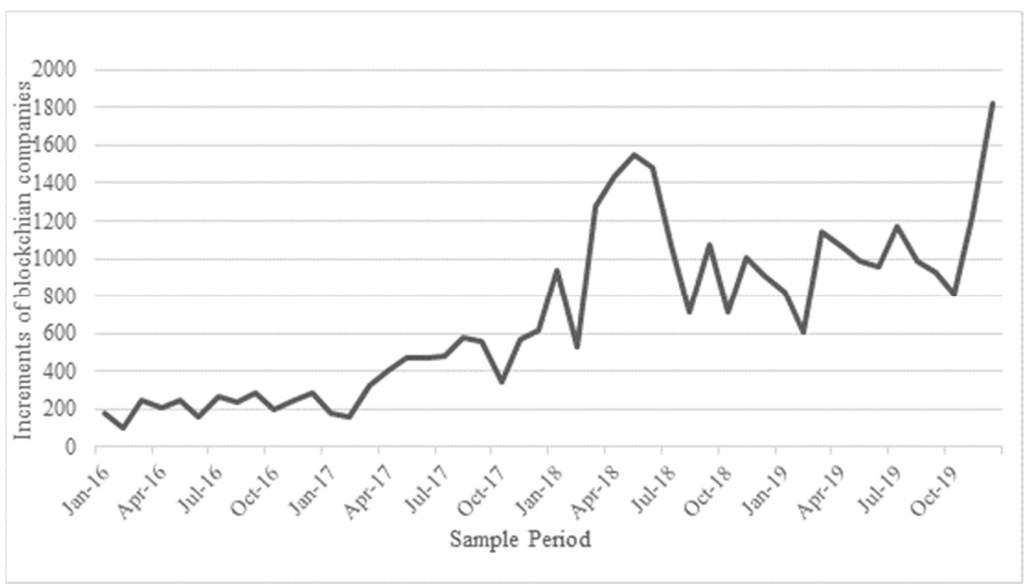

**Figure 1.** The increments of new blockchain companies registered by website (Figure 1 shows the number of newly registered blockchain companies per month in 2016–2019. From 2016 to 2018, the number of registered blockchain companies fluctuated but increased. However, in 2019, digital currency was affected by the stock market turmoil and continued to decline, which also affected the development of blockchain enterprises).

However, the relationship between new technologies and productivity has long been debated [10–12]. Solow (1987) [13] surveyed 292 companies in the late 1980s and noted that there was no clear correlation between investment and return on investment. In a recent study, Acemoglu et al. (2014) [11] found that it is difficult to square output declines with the notion that computerization and IT embodied in new equipment are driving a productivity revolution, at least in US manufacturing. Toward this, we have to reconsider the important role of blockchain technology and its relationship to production efficiency.

Unfortunately, it has become a research gap to evaluate the performance of blockchain technology, especially using a standardized large sample. This may be due to the complexity of measurement for blockchain development. Some studies have tracked companies that announced on social media that their blockchain development projects had completed operational deliveries and found that these companies created hype to achieve rapid stock price gains [14–16]. This signals to us that it is possible for companies to falsely claim to be involved in blockchain technology in pursuit of short-term profits.

Inspired by Mukim (2014) [17], this paper counts the number of blockchain companies in the city where the listed company is located to construct a blockchain development indicator. Our basis is that listed companies are more inclined to carry out technical cooperation with surrounding companies, so the more blockchain companies in the city, the more likely they are to use blockchain technology. In the robustness test, on the basis of same assumption, we recorded the detailed latitude and longitude of listed companies and blockchain companies and calculated the number of blockchain companies within a 30 km around the listed company as a proxy variable for blockchain development. This paper presents new evidence that blockchain technology can improve the TFP of enterprises by reconstructing the production process of enterprises and improving the value network of enterprises.

The paper contributes to the previous literature in three main ways. First, it adds to the growing body of blockchain literature. To the best of our knowledge, our study is the first to examine the impact of blockchain technology on firm TFP from an empirical perspective. Gausdal et al. (2018) [18] conducted a case study on the adoption of blockchain technology in the Norwegian offshore industry, with data collected from interviews. Sim-

ilarly, Davidson (2016) [19] also adopted the case study method on the Ethereum-based infrastructure protocol and platform backfeed. In this paper, we included the data of Chinese A-listed companies during 2017–2019 as samples, and manually collected detail addresses of blockchain companies, which expanded the research methods.

Second, our work also contributes to the construction of blockchain development indicators. Queiroz et al. (2019) [20] constructed a model developed in accordance with prior IT adoption literature. However, this model did not involve the application of blockchain technology and could reflect the real situation on a narrow scale. Another approach is to track news reports and company announcements which announce the incorporation of blockchain technology [14–16]. We acknowledge that the practice is reasonable. The main problem, however, is that companies often have little intention to develop blockchain technology, but to take advantage of the popularity of this technology among investors.

In our analysis, to overcome the limitations, we obtained detailed registration information of 68,842 blockchain companies and constructed an indicator by using the number of blockchain companies in each city. Simultaneously, for accuracy, we computed the latitude and longitude of listed companies and blockchain companies and counted the number of blockchain companies within 30 km around the listed company. Prior research on blockchain technology development has mainly been based on qualitative analysis and focused on the behavioural consequences [18]. We contribute to the literature by constructing a novel proxy indicator, which could be objective to avoid the profit-seeking behaviour of catching hot information. Moreover, the indicator based on registration information can alleviate the endogeneity problem and provide a useful reference for the subsequent construction of blockchain indicators.

Third, this paper provides evidence for the Productivity Parado. In several studies, technological progress was considered to be the driving force for the improvement of TFP. However, follow-up studies suggest that the impact of digital technologies on TFP is not obvious [21,22]. After the rise of a new generation of technological revolution, this debate still exists and our research provides new empirical evidence for the positive effects of digital technologies.

The structure of the paper is as follows: Section 2 offers how the study relates to the literature and proposes hypotheses. Section 3 introduces the methodology and data. Section 4 presents and discusses the test results. Section 5 concludes this study and provides suggestions for future study.

## 2. Literature and Hypothesis

### 2.1. Digital Economy and Firm Total Factor Productivity

The development of blockchain technology is a part of the digital technology. In general, human society is undergoing a process of digitization, and blockchain technology plays an important role in facilitating this process [23]. Existing research on the digital economy and productivity can be divided into two aspects. The first investigates the impact of the local digital economy industry on productivity. Jung et al. (2014) [24] studied the interstate data in Brazil from 2007 to 2011 and found that less economically advanced regions usually have problems such as insufficient factor endowment and lack of natural resources, and the Internet provides a new important resource for the economic development of these regions. Chu (2013) [25] surveyed data from 201 countries, empirically finding that a 10% increase in Internet penetration can increase per capita GDP by 0.57% to 0.63%.

The second studies the impact of digital transformation on firm productivity. Research in this area mainly includes the impact of finance and artificial intelligence applications [19,26–28]. However, conclusions in research are divided on this issue. Dahl et al. (2017) [29] believe that banks' high investment in network security and big data processing drags down their performance. On the contrary, Fuster et al. (2019) [30] used U.S. mortgage loan data and found that fintech banks process mortgage applications 20% faster than other banks, and

that faster processing does not come at the cost of high default rates. Huang et al. (2018) [31] reported that financial technology could energize traditional financial institutions through technology, which can alleviate the information asymmetry between traditional financial institutions and enterprises in many ways and improve the efficiency of credit allocation.

## 2.2. Blockchain Technology and Firm Total Factor Productivity

According to the definition issued by the Ministry of Industry and Information Technology of China ("*China's Blockchain Technology and Application Development White Paper*", issued by the Ministry of Industry and Information Technology of China in 2016), blockchain is a new application model of computer technology such as distributed data storage, point-to-point transmission, consensus mechanisms, and encryption algorithms. As one of the most promising technologies, blockchain technology can achieve decentralization and reduce intermediary costs. Its traceability and immutability can effectively solve data protection and information sharing problems, rebuild the organizational form of enterprises, and improve the productivity of firms.

The endogenous growth theory posits that the main driving force of economic development comes from innovation activities, and the production and dissemination of information technology are particularly important for the accumulation of knowledge. One of the technical advantages of the blockchain lies in the processing, integration, and sharing of information [19,32]. Each node on the chain can share existing information and reprocess it. The distributed and traceable data storage method not only breaks through the limitation of time and space but also creates a reliable cooperation mechanism for the dissemination of information resources. Autio (2018) [33] believes that digital technology represented by blockchain features decoupling, disintermediation, addressability, and memory. These characteristics help to reduce the reliance on intermediary and proprietary elements of the value chain in the production process, thereby significantly reducing transaction costs and greatly improving productivity.

According to the empowerment theory, digital technology can be combined with traditional production factors and reconstruct the production factor system, thereby providing a series of value-added services for the industrial chain. Blockchain technology mainly empowers the real economy from three aspects. First, blockchain technology is conducive to the further development of the Industrial Internet and the deep integration of informatization and industrialization [34]. Second, blockchain technology can promote the coordinated development of industrial chains [35]. Third, blockchain technology can reshape the enterprise value chain process, such as the application of blockchain technology in finance, which has reconstructed the current transaction rules, payment and settlement systems, and regulatory systems. The smart contracts and consensus mechanisms effectively solve the problems of data security and trust risks.

Under the framework of the endogenous growth theory and empowerment theory, blockchain technology can drive the improvement of the TFP of enterprises through "Technical Channels" and "Efficiency Channels". "Technology Channels" refers to the reconstruction of the firm production process by blockchain technology. For listed companies, an important channel is the development of the double-chain integration model of blockchain and supply chain [36]. Relying on smart contract technology, firms can realize the automatic operation of the overall process of the supply chain; the traceability of blockchain technology enables information sharing among foundries, suppliers, enterprises, and agents, establishing a reliable trust mechanism. "Efficiency Channels" refers to the empowerment of blockchain technology to firms, alleviating the problem of information asymmetry and reducing knowledge search costs, management costs, and operating costs, thereby promoting enterprise upgrading and improving TFP. Accordingly, the following is proposed:

**Hypothesis 1 (H1).** *Blockchain technology promotes the TFP growth.*

## 3. Methodology and Data

### 3.1. Empirical Models

To assess the impact of blockchain development on TFP growth, we built the following model using the ordinary least square (OLS) method:

$$\ln TFP_{cift} = \alpha_0 + \alpha_1 \ln Blockdev_{ct} + \beta X_{ct} + \gamma F_{it} + \delta_i + \delta_{ft} + \mu_{cift} \tag{1}$$

where $TFP_{cift}$ denotes the total factor productivity of enterprise $i$ in city $c$ of province $f$ in year $t$. $Blockdev_{ct}$ denotes blockchain development, constructed as the natural logarithm of one plus the number of blockchain enterprises in city $c$ in year $t$. $X_{ct}$ contains the city-level control variables; $F_{it}$ denotes the firm-level control variables that change with time; $\delta_i$ represents the individual fixed effects of firms that do not change with time. Due to the heterogeneity of economic policies and economic development in different provinces, this paper also controls for the interactive fixed effects of province and year. Table 1 shows the specific variable definitions.

**Table 1.** Definitions of variables.

| Variable Name | Variable Label |
|---|---|
| Panel A: Dependent variables | |
| TFP_OP | Firm $i$'s Total Factor Productivity calculated by OP method during year $t$. |
| TFP_LP | Firm $i$'s Total Factor Productivity calculated by LP method during year $t$. |
| Panel B: Independent variables | |
| Blockdev | Logarithm of (1 + no. of blockchain companies in city). |
| Blocknum | Logarithm of (1 + no. of blockchain companies within 30 km of listed company). |
| Panel B: control variables | |
| Finance | Loan balance of financial institutions/Gross Domestic Product of city where firm $i$ is located |
| Internet | Logarithm of (1 + no. of Internet users in city). |
| Perdensity | The population density of city where firm $i$ is located. |
| Pergdp | The per-capita GDP of city where firm $i$ is located. |
| ROE | Firm $i$'s return on equity, which equals to net income divided by total assets during year $t$. |
| Tobin's Q | Firm $i$'s ratio of the sum of market value of equity plus book value of debt to book value of assets at year $t$. |
| Level | Firm $i$'s book value of total debts divided by the book value of total assets during year $t$. |
| Stockowner | Firm $i$'s shareholding ratio of the largest shareholder. |
| Total asset | Firm $i$'s total asset growth rate. |
| Size | Logarithm of (Total Assets). |
| Director | Firm i's independent directors percentage on the board of directors. |

### 3.2. Data Collection

We collect data from various sources. Within the blockchain development indicators, we obtained blockchain enterprise information from the "Qichacha" website (https://www.qcc.com/, accessed on 4 March 2021). The data on Qichacha comes from the State Administration for Industry and Commerce of the People's Republic of China and includes industrial and commercial information, investors, foreign investment, corporate annual reports, lawsuits, untrustworthy information, patents, copyrights, trademark displays, corporate certificates, corporate news, corporate recruitment, and other information. We excluded some companies with abnormal operating conditions. The regional data was extracted from The Statistical Yearbook of Prefecture-level Cities (2017–2020). All firm-level information, including financial data and control variables, were collected from the China Stock Market & Accounting Research Database (CSMAR), a comprehensive research-oriented database focusing on China Finance and Economy. Although the time span of the article is relatively short, it is representative and completely reflects the development process of blockchain. In 2016, China's industrial information and informatization

department included blockchain in the national development plan, representing the promotion of blockchain technology development. From 2017 to 2018, China has successively introduced regulatory policies for "blockchain technology" projects to remove bubbles, such as cracking down on "false" behaviors. Until 2019, blockchain regulatory policy was gradually maturing and completing, and the core technology was developing steadily. Our data covers up to 2020, as the emergence of the COVID-19 pandemic has disrupted the registration applications of blockchain businesses.

### 3.3. Variable Measurement

Firm Total Factor Productivity (TFP). Before estimating total factor productivity, it is usually necessary to subscribe to the form of the production function. In the existing literature, the Cobb–Douglas production function (C–D production function) has become the most frequently cited functional form, usually in the following form:

$$Y_{it} = A_{it} L_{it}^{\alpha} K_{it}^{\beta} \tag{2}$$

However, due to the correlation between productivity shocks and the probability of firms exiting the market, the estimated TFP will have problems such as sample selection bias. Olley and Pakes (1996) [37] developed a method based on the consistent semi-parametric estimator (OP method). This method assumes that firms make investment decisions based on the current state of firm productivity; so, the firm's current investment is used as a proxy variable for unobservable productivity shocks, thereby solving the problem of simultaneity bias.

In this paper, according to the idea of the OP method, export behavior decision-making is introduced into the OP framework and the following model is constructed:

$$\ln Y_{i,t} = \beta_0 + \beta_k \ln K_{i,t} + \beta_l \ln L_{i,t} + \beta_m M_{i,t} + \beta_\alpha Age_{i,t} + \\ \beta_s SOE_{i,t+} \beta_e Export_{i,t} + \sum \gamma_m Year_m + \sum \delta_n Industry_n + \sum \xi_k prov_k + \varepsilon_{it} \tag{3}$$

where $Y$ denotes the output of the firm, measured by the main business income; $K$ denotes the capital stock, constructed by the sum of original value of fixed assets, engineering materials, and construction in progress; $L$ denotes the labor input, expressed by the number of employees in the enterprise; and $M$ denotes corporate investment. $Age$, $SOE$, and $Export$ denote the age of the enterprise, state-owned enterprise dummy variable, and overseas income dummy variable, respectively, which represent the individual characteristics of the enterprise. The subscripts $i$ and $t$ represent firm and time, respectively. Under the condition of panel data, we introduce industry dummy variables, province dummy variables, and time dummy variables to solve the possible endogeneity problem and obtain a consistent and unbiased estimate of the production function.

According to the definition of TFP, it can be seen that: $\ln TFP_{it} = \beta_0 + \varepsilon_{it}$, thus the absolute level value of TFP can be obtained: $\ln TFP_{it} = \ln Y_{it} - \beta_k \ln K_{it} - \beta_l \ln L_{it}$.

*Blockchain development level.* Referring to Ye et al. (2019) [38], this article constructs blockchain development using the number of blockchain companies owned by the city where the listed company is located. According to Mukim (2014) [17], in areas with more blockchain companies, the more likely it is that listed companies will reach strategic cooperation with blockchain companies. Specifically, we first obtained registration information from the website and screened out the samples containing the feature word. Then, in order to alleviate coincidental information, we extracted simples that include feature words in "*business name*" and "*business scope*", and manually removed companies that did not actually develop blockchain technology; we also excluded some industries that do not rely on production as their main business, such as the financial industry and the advertising industry. Finally, we excluded some samples with abnormal operations, such as cessation of business, dissolution, or revocation. The final sample includes 68,842 observations (as of 31 December 2020), which is close to the data published by the *Blockchain Homepage*

(https://bc.cert.org.cn/, accessed on 4 March 2021). According to the enterprise address, we obtain the number of blockchain enterprises owned by each city every year.

In the robustness test, the number of blockchain enterprises within 30 km of the listed company is used as an alternative indicator for blockchain development. Here, we collected the registered address of the listed company and the registered address of the blockchain enterprises first, and used the map to convert the registered address into latitude and longitude coordinates. Second, based on the obtained corporate address coordinates, we calculated the geographic distance between the listed company and all blockchain companies, Third, we kept blockchain companies within 30 km of the listed company, and counted the number of companies.

*Control variables.* Following the established literature [14,39], we controlled the factors that might affect the enterprises at the individual and city level. At the firm level, we controlled the ratio of Asset liability, Tobin's Q, Growth rate of Total Assets, ratio of largest shareholder, and ratio of independent directors. At the city level, we controlled the local number of Internet users, level of financial development, population density, and per-capita GDP of the city. Table 1 presents summary statistics for the variables used in the analysis.

### 3.4. Summary Statistics

Table 2 reports the descriptive statistics for the variables used in the following analysis with a total of 8772 firm-year observations. On average, the total factor productivity of listed companies is 2.303 and the standard deviation is 0.127. The average number of blockchain enterprises owned by cities in China is about 22 and the standard deviation reaches the fluctuation range of 9 enterprises, reflecting the variability of the data. The average number of blockchain companies within 30 km around the registered place of listed companies is about 7, which shows great volatility.

**Table 2.** Descriptive statistics *.

| Variables | Obs. | Mean | SD | Min | Max |
|---|---|---|---|---|---|
| lnTFP_OP | 8772 | 2.303 | 0.127 | 2.167 | 2.895 |
| lnTFP_LP | 8772 | 2.435 | 0.194 | 2.183 | 2.934 |
| Blockdev | 8772 | 3.069 | 2.167 | 0 | 8.611 |
| Blocknum | 8772 | 1.955 | 1.946 | 0 | 5.358 |
| Finance | 8772 | 0.554 | 0.876 | 0 | 2.682 |
| Internet | 8772 | 0.541 | 0.241 | 0.152 | 1.112 |
| Perdensity | 8772 | 6.644 | 0.752 | 4.262 | 7.923 |
| Pergdp | 8772 | 10.967 | 13.418 | 0 | 53.235 |
| ROE | 8772 | 0.062 | 0.138 | −0.829 | 0.327 |
| Tobin's Q | 8772 | 1.782 | 0.969 | 0.841 | 6.731 |
| Level | 8772 | 0.413 | 0.196 | 0.064 | 0.879 |
| Stockowner | 8772 | 0.336 | 0.144 | 0.085 | 0.724 |
| Total asset | 8772 | 0.141 | 0.277 | −0.342 | 1.785 |
| Size | 8772 | 21.629 | 1.776 | 17.928 | 25.405 |
| Director | 8772 | 0.378 | 0.054 | 0.333 | 0.571 |

* This table reports the summary statistics for the variables listed in Table 1. The sample period is 2016–2019.

## 4. Results

### 4.1. Basic Regression Results

Table 3 presents the regression results of the blockchain development on the individual TFP. The paper added the control variables gradually to observe the difference in the key coefficients. Column (1) controls the individual effect and the interactive fixed effects of province and year only. The estimated coefficient of the blockchain development level is 0.037, which is significant at the 1% level. To avoid the influences of variables that affect both enterprise productivity and the level of blockchain development in cities, columns (2)–(4) gradually control the variables at the regional level, including the number of local Internet users (Internet), local financial development (Finance), population density (Perdensity), and

urban GDP per capita (Pergdp). Columns (6)–(10) report the results for adding firm-level control variables, including Asset liability, Tobin's Q, Growth rate of Total Assets, ratio of largest shareholder, and ratio of independent directors, and the coefficients are significant and tend to be stable. Table 1 shows the specific variable definitions.

**Table 3.** Basic regression results *.

| Variables | (1) | (2) | (3) | (4) | (5) | (6) |
|---|---|---|---|---|---|---|
| Blockdev | 0.037 *** | 0.027 *** | 0.022 *** | 0.021 *** | 0.019 *** | 0.019 *** |
| | (0.002) | (0.003) | (0.003) | (0.003) | (0.003) | (0.003) |
| Internet | | 0.145 *** | 0.085 *** | 0.086 *** | 0.076 ** | 0.070 ** |
| | | (0.029) | (0.030) | (0.031) | (0.034) | (0.034) |
| Finance | | | 0.028 *** | 0.031 *** | 0.036 *** | 0.037 *** |
| | | | (0.004) | (0.006) | (0.007) | (0.007) |
| Perdensity | | | | 0.049 | 0.079 | 0.076 |
| | | | | (0.061) | (0.062) | (0.061) |
| Pergdp | | | | 0.001 * | 0.001 * | 0.001 * |
| | | | | (0.001) | (0.001) | (0.001) |
| Tobin's Q | | | | | 0.013 * | 0.016 ** |
| | | | | | (0.007) | (0.007) |
| Total asset | | | | | 0.006 | −0.039 *** |
| | | | | | (0.014) | (0.014) |
| Size | | | | | 0.049 *** | 0.031 *** |
| | | | | | (0.013) | (0.011) |
| Level | | | | | | 0.181 *** |
| | | | | | | (0.064) |
| Stockowner | | | | | | −0.002 |
| | | | | | | (0.001) |
| ROE | | | | | | 0.363 *** |
| | | | | | | (0.031) |
| Director | | | | | | 0.211 |
| | | | | | | (0.130) |
| Cons | 7.720 *** | 6.912 *** | 7.267 *** | 7.582 *** | 7.919 *** | 7.834 *** |
| | (0.007) | (0.159) | (0.170) | (0.425) | (0.434) | (0.427) |
| Firm FE | Yes | Yes | Yes | Yes | Yes | Yes |
| Province × Year FE | Yes | Yes | Yes | Yes | Yes | Yes |
| N | 8772 | 8772 | 8772 | 8772 | 8772 | 8772 |
| $R^2$ | 0.051 | 0.055 | 0.064 | 0.064 | 0.067 | 0.099 |

* This table reports the results (standard errors in parentheses) of blockchain development on firms' TFP under different control variables. The dependent variable is blockchain development, measured as the natural logarithm of one plus the number of blockchain enterprises in cities, the independent variable is firms' TFP. In column (1), the paper controls the individual effect and interactive effect of province and time. Columns (2)–(4) gradually control the variables at the regional level (Internet, Finance, Perdensity, Pergdp), columns (6)–(10) report the results for adding firm-level control variables (Internet, Finance, Perdensity, Pergdp). *, **, and *** indicate statistical significance at the 10%, 5%, and 1% levels, respectively.

In the above analysis, this paper has largely controlled the effect of the omitted variables, and the interactive fixed effects of province and time can effectively limit the impact of those unobservable omitted variables that change over time. Thus, the results strongly support our conjecture that the higher the level of blockchain development, the higher the level of TFP of enterprises.

*4.2. Endogeneity Concerns*

Enterprises with high productivity are likely to be more active in developing or participating in blockchain-related business toward improving productivity with the help of the technical advantages of blockchain. Therefore, this paper considers the possible reverse causality problem.

This paper introduces the instrumental variable method to solve the problem of reverse causality. Precisely, we construct the blockchain density index, that is, the proportion of the

number of blockchain policies issued by a province (and municipality directly under the Central Government) to the total number of policies in the country each year.

In terms of relevance, because of China's market economy policy, the number of blockchain-related policies in a province can directly affect the level of blockchain development in that province. From the exogenous point of view, policies will not directly improve firms' TFP, and the productivity of enterprises cannot directly affect the formulation of policies.

However, this indicator has the following two problems. First, the explanatory variables are at the prefecture-level city level, while the constructed instrumental variables are at the provincial level. This indicator cannot fully reflect the heterogeneity of blockchain development in each city. Second, from a practical standpoint, China's regional economic development shows a phenomenon of resource agglomeration [40], radiating from the provincial capital (municipal) as the center, which means that a provincial capital city will have more blockchain companies. (China's political system is roughly composed of five national administrative layers: central (central), province (province), prefecture (region), county (county), and township (township) (Li and Zhou, 2005). Provinces are the second level of China's political hierarchy, playing a very important role in economic management (Qian and Xu, 1993). The number of policy documents reflects the attention of government authorities to blockchain, highly correlated with the likelihood of blockchain technology adopted by firms.)

Therefore, this paper further uses the logarithm of the distance from the prefecture-level city to its provincial capital city as the weight to correct the provincial blockchain density index. The formula is as follows:

$$pcblock_{ct} = \frac{policy_{pt} / \sum_{p=1}^{31} policy_{pt}}{\ln dis \tan ce} \tag{4}$$

where the numerator is the provincial blockchain density index, the denominator is the distance weight, and the weight of the provincial city and the municipality itself is 1.

Table 4 presents the estimated results using this instrumental variable. Columns (1) and (3) do not control for fixed effects, while columns (2) and (4) further control the estimated results for individual effects. In both cases, the F-values for the weak correlation test were greater than the critical level of 10, and the second-stage estimates were significant at the 1% level.

**Table 4.** Results of instrument variable analysis *.

| Variables | The First Stage | | The Second Stage | |
|---|---|---|---|---|
| | **(1)** | **(2)** | **(3)** | **(4)** |
| pcblock | 0.041 *** | 0.033 *** | | |
| | (0.001) | (0.002) | | |
| blockdev | | | 0.021 *** | 0.020 *** |
| | | | (0.003) | (0.003) |
| Controls | Yes | Yes | Yes | Yes |
| Firm FE | No | Yes | No | Yes |
| Province × Year FE | No | Yes | No | Yes |
| *F*-Vlaue | 16.462 | 38.576 | | |
| *N* | 8536 | 8630 | 8536 | 8630 |
| $R^2$ | 0.046 | 0.048 | 0.064 | 0.098 |

* The table reports the results of the 2SLS estimation using instrument variables. The independent variable in columns (1) and (2) is pcblock, which represents the number of policies related to the blockchain. Columns (2) and (4) are the regression result of the second stage. In columns (1) and (3), we do not control the firm and interaction of province and time fixed effect. In columns (2) and (4), we control all the variables. Construction and definitions of the variables are provided in Table 1. The table reports coefficient estimates followed by robust standard errors. All models include a full set of control variables, and the results are available on request from the authors. *** indicates statistical significance at 1% levels.

*4.3. Heterogeneity Analysis*

So far, our results indicate that blockchain development has a significant positive impact on firm TFP. In this section, we focused on whether the favorable outcomes of blockchain development are influenced by the enterprise features, such as firm ownership, the industry the firm operates in, and the firm's initial productivity. The following analysis introduces the number of blockchain companies owned by the city where the company is located to investigate the influence of the factors above. (We also used the number of blockchain companies within 30 km around the enterprise for verification. The result is still valid, but it is not presented in the paper. Please contact the corresponding author for the result.)

4.3.1. Firm Ownership, Blockchain Development, and Total Factor Productivity

The paper decomposes the firm sample by SOEs, non-SOEs, and estimate models (1). Columns (1) and (2) in Table 5 report the regression results for the SOE and non-SOE samples, respectively. The results show that, for the SOEs, the coefficient of blockchain development is positive but not significant, while for non-SOEs, the coefficient of blockchain development is significantly positive. The reason for this may be that in the usage of new technologies, although SOEs have advantages in resources, they lack an effective conversion mechanism [41,42]. In contrast, non-SOEs are sensitive to the market of digital technology, and flexible in transformation, cooperation, and digital technology construction [41]; so, they can convert technical elements into productivity more efficiently than SOEs.

**Table 5.** Heterogeneity analysis *.

| | (1) | (2) | (3) | (4) | (5) | (6) |
|---|---|---|---|---|---|---|
| | **SOEs** | **Non-SOEs** | **Surplus Ind** | **Non-Surplus Ind** | **High Initial Productivity** | **Low Initial Productivity** |
| Blockdev | 0.002 | 0.024 *** | 0.019 | 0.025 *** | 0.015 *** | −0.001 |
| | (0.008) | (0.009) | (0.027) | (0.009) | (0.006) | (0.008) |
| Cons | 7.655 *** | 7.800 *** | 6.161 | 7.115 *** | 8.764 *** | 6.662 *** |
| | (0.708) | (0.492) | (5.182) | (1.425) | (0.531) | (1.253) |
| Controls | Yes | Yes | Yes | Yes | Yes | Yes |
| Firm FE | Yes | Yes | Yes | Yes | Yes | Yes |
| Province × Year FE | Yes | Yes | Yes | Yes | Yes | Yes |
| N | 2893 | 3715 | 3212 | 3715 | 1916 | 1493 |
| $R^2$ | 0.121 | 0.084 | 0.143 | 0.128 | 0.073 | 0.086 |

* This table presents the heterogeneity analysis for the main results (standard errors in parentheses). The dependent variables are all blockchain development and the independent variables are all firms' TFP. For brevity, we denote all control variables as controls, and the control variables are the same as those in Table 3. In columns (1) and (2), we construct a dummy variable that indicates whether the firm is an SOE to divide the sample into two parts. In columns (3) and (4), we divide the full sample into excess capacity industries and non-excess capacity industries. According to the classification standards announced by the Ministry of Commerce of China, industries with excess capacity include steel, building materials and housing construction, mineral resource development, petrochemical, natural gas energy, and other industries. In columns (5) and (6), we limited the sample to manufacturing enterprises. Column (5) shows the results of blockchain influences on the TFP of firms with high initial productivity and column (6) shows the results of blockchain influences on the TFP of firms with low initial productivity. All models include a full set of control variables, and the results are available on request from the authors. *** indicates statistical significance at the 1% levels.

4.3.2. Industry, Blockchain Development, and Total Factor Productivity

The application of new technology is not only influenced by the management model and institutional culture but is also closely related to the industry in which it is located. Liu et al. (2017) [43] believe that the profits of industries with excess capacity are affected by the macroeconomic cycle, and they are conservative in business strategies. This paper divided the industries in which listed companies are located into industries with excess capacity (surplus Ind) and industries without excess capacity (non-surplus Ind), and then performed group regression. The results are shown in columns (3) and (4) of Table 5. The impact of the development of blockchain technology on firms' TFP is not significant in

the excess capacity industry (surplus Ind), but it is significantly positive in the non-excess capacity industry (non-surplus Ind). The findings are consistent with Liu et al. (2017) [43]. In industries with excess capacity, such as the steel, petroleum, petrochemical, and other traditional industries, the survival and development of enterprises are often under pressure, which makes it difficult for enterprises to apply new technologies.

### 4.3.3. Initial Productivity, Blockchain Development, and Total Factor Productivity

The influence of blockchain development on firm productivity may be affected by the initial productivity of the enterprise, that is, compared with companies with lower initial productivity, companies with high initial productivity can take full advantage of the technological effects introduced by digitization. To gain a clearer understanding of this inference, this paper limited the sample to manufacturing enterprises and investigated the impact of blockchain development on enterprise productivity under different initial productivity levels. To alleviate the bias of the single-year measurement, this paper limits the samples to those from 2014 to 2015, takes the mean value of initial productivity as the standard, and selects the 75% and 25% quartile samples for regression. Columns (5) and (6) of Table 5 report the estimated results. The results show that, for enterprises with high initial productivity, the impact of blockchain development on enterprise productivity is significantly positive, while, for enterprises with low initial productivity, the estimated coefficient of blockchain development is negative and insignificant. The results confirm our inference.

### 4.4. Robustness Checks

In this section, we provide robustness checks to confirm our main findings, and our results are robust to a variety of identifications.

First, to investigate whether there may be measurement errors with our key variable, we examined our conclusion for alternative proxies. Following Du et al. (2014) [44], this paper identifies the longitude and latitude of the listed company's registered addresses, obtains the distance between the registered address of each listed company and the blockchain company's registered address, and, finally, counts the number of blockchain companies within 30 km of each listed company. The estimation results of column (1) in Table 6 show that the coefficient estimates are significantly positive at the 1% level.

Second, as an alternative measure of TFP, we further used the Levinsohn–Petrin method (LP method, [45]) to estimate TFP. Instead of using the investment value, the LP method uses the price of the intermediate input as a proxy variable. Column (2) provides the results; after controlling for individual effects, the interactive fixed effects of province and year, and the basis for controlling variables, the coefficient of the blockchain development level is still significantly positive at the 1% level.

Third, we eliminated the influence of biased samples. In columns (3)−(5), we restricted the samples whose registered addresses were in big cities, companies with unstable development (established for less than one year), and companies whose main business focuses on technology. We found that (i) Geographically, economically developed regions have higher productivity and are more likely to enlist the help of blockchain technology. To eliminate the confounding effect caused by geographical location, this paper further eliminates the sample of provincial capital cities and municipalities directly under the Central Government. Column (3) reports the estimation results and the results are robust; (ii) It takes time for a blockchain enterprise to actually conduct business with a listed company. Therefore, based on the industrial and commercial registration information of blockchain companies, this article excludes blockchain companies that are too young (established for less than one year). The results are shown in column (4); although the estimated coefficient has decreased, it is still significantly positive at the 5% level, indicating that the results are reliable. (iii) Most of the leading blockchain businesses in China are launched first in the technology industry; so, we have to consider the sample self-selection bias. Column (5)

shows the results of excluding the sample of listed companies in the technology industry and the result is still significantly positive at the 5% level.

**Table 6.** Robustness checks *.

| Variables | (1) | (2) | (3) | (4) | (5) | (6) |
|---|---|---|---|---|---|---|
| | **lnTFP_OP** | **lnTFP_LP** | **lnTFP_OP** | **lnTFP_OP** | **lnTFP_OP** | **lnTFP_OP** |
| Blockdev | | 0.025 *** | 0.023 *** | 0.011 ** | 0.016 ** | 0.022 ** |
| | | (0.005) | (0.007) | (0.005) | (0.006) | (0.010) |
| Blocknum | 0.036 *** | | | | | |
| | (0.006) | | | | | |
| L. Blockdev | | | | | | 0.004 ** |
| | | | | | | (0.002) |
| L2. Blockdev | | | | | | 0.009 * |
| | | | | | | (0.005) |
| Cons | 7.506 *** | 10.886 *** | 8.357 *** | 8.778 *** | 8.046 *** | 7.764 *** |
| | (0.281) | (0.223) | (0.359) | (0.318) | (0.265) | (0.233) |
| Controls | Yes | Yes | Yes | Yes | Yes | Yes |
| Firm FE | Yes | Yes | Yes | Yes | Yes | Yes |
| Province $\times$ Year FE | Yes | Yes | Yes | Yes | Yes | Yes |
| $N$ | 8652 | 8772 | 4878 | 4176 | 5482 | 5416 |
| $R^2$ | 0.093 | 0.118 | 0.053 | 0.072 | 0.055 | 0.082 |

* This table presents the robustness checks for the main results (standard errors in parentheses). The dependent variable is blockchain development and the independent variable is firms' TFP. For brevity, we denote all control variables as controls, and the control variables are the same as those in Table 3. In column (1), we employ the number of blockchain companies within 30 km around the listed company as the alternative proxy for measuring blockchain development. In column (2), the paper applies TFP measured by the LP method as the alternative proxy for firms' TFP. In columns (3)−(5), we restrict the samples whose registered address is in big cities, companies with unstable development (established for less than one year), and companies whose main business focuses on technology. In column (6), we lag the effect of blockchain development into two periods, respectively. All estimations control for firm and the interaction of province and time. All models include a full set of control variables, and the results are available on request from the authors. *, **, and *** indicate statistical significance at the 10%, 5%, and 1% levels, respectively.

Fourth, we decompose the effect of blockchain development into different time periods to examine the dynamic effects of blockchain technology on a firm's TFP. In column (6), we lag the blockchain indictor by 1 and 2 years, respectively. The results show that the coefficient of the blockchain indicator is significant at the level of 5% when the lag is one period; with a lag of two periods, the coefficient is still significant. In sum, the impact of blockchain technology on TFP is sustained for approximately two years.

## 5. Conclusions

As the digital production factor with the most potential, blockchain technology is of great significance for digital transformation and the optimization of factor allocation [46]. The paper creatively constructs the blockchain development level using the number of local blockchain companies to investigate the impact of regional blockchain development on the TFP of listed companies, and the regression result validates the hypothesis. Considering possible endogeneity issues, this paper combines exogenous policies and geographic distances to construct instrumental variables to effectively identify causal effects. The analysis found that the regional development of blockchain technology can significantly promote the improvement of the firms' TFP. After robustness tests, the conclusion is still positive. Additionally, this paper further examines the relationship between the impact of blockchain technology in a different sample. First, we find that the sample of state-owned enterprises is not statistically significant; second, the positive influence is more significant in the non-overcapacity industry sample; third, firms with high initial productivity use blockchain technology to increase productivity more obviously.

From this paper, there is sufficient evidence that the development of blockchain technology can significantly improve the TFP of enterprises, which should interest regulators

and investors. As a revolutionary technology, it is significant to promote the in-depth development and accelerate the implementation. For enterprises, the development of blockchain technology should consider cost and efficiency. In addition to the initial investment in technology, resources, and fixed assets, with the operation of the blockchain network, the operation and maintenance cost rises accordingly. Especially for enterprises that are in traditional industries with low economic benefits, it is necessary to fully study and judge the cost and efficiency of the application of new technologies.

Regarding the main shortfalls of this work, one of them may be the limited available data. Our span of panel time only includes from 2016 to 2019, which limits the comparison of results. Another limitation is that this study was conducted in a single country. That notwithstanding, it gives room for future studies, which could apply and extend our indictors and model in other countries or other aspects. For example, they could compare an emerging country with a developed country in terms of results and findings. They could also conduct specific research on the application of blockchain technology, such as in the supply chain and fintech.

**Author Contributions:** Q.C. drafted the manuscript and performed the statistical analysis. J.L. conceived of the study and participated in its design and coordination and helped to draft the manuscript. H.Z. participated in the design of the study and performed the statistical analysis. Y.L. and X.L. organized data. All authors have read and agreed to the published version of the manuscript.

**Funding:** This research was funded by the Major Program of National Social Science Foundation of China, grant number 20&ZD128, the Key Program of National Social Science Foundation of China, grant number 20ASH008, and Postgraduate Research & Practice Innovation Program of Jiangsu Province, grant number KYCX22_2984.

**Institutional Review Board Statement:** Not applicable.

**Informed Consent Statement:** Not applicable.

**Data Availability Statement:** Data are available upon request due to restrictions, e.g., privacy or ethics. The data presented in this study are available upon request from the corresponding author.

**Conflicts of Interest:** The authors declare no conflict of interest.

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
