# Peer review of "Blockchain and Firm Total Factor Productivity: Evidence from China"

_sustainability, doi:10.3390/su141610165_

Round 1

Reviewer 1 Report

The topic of the article is very current, it reflects the current problem, when, for example, due to high energy prices, even digital services can be limited and outages can occur.

The article presents the results of 8,772 observations for the years 2016 to 2019. In the abstract of the article it is written that the authors creatively used constructs blockchain development indicators using geographical 14 characteristic. In the article itself, this model is used, however, the description of the "creative solution and contribution of the authors" disappears - Is it possible to enlarge this part?
Furthermore, I am missing a discussion in the article - how do your results differ from the studies described in chapter 2.1 and 2.2? At the end of the article, only a summary of your results and recommendations is made.

Please revise - Figure 1.is too small and no source of data in the graph of blockchains.

Reviewer 2 Report

The Figure 1 is too small and it is hard to read. The content of the theoretical background can be improved. The Conclusion can be expanded. Also  English language and style can be spell checked.

Reviewer 3 Report

  • The paper investigates the effects of blockchain application on Chinese firms productivity. The work is sufficiently clear in its structure and it presents a coherent theoretical background. GAP of the literature and research questions are both well identified, as well as the results that derive from. 
    The methodology has some shortcomings. As outlined by the authors in the research limitations section, the data collection only covered the period 2016 - 2019 and it is only referred to the Chinese market. These factors limit the research results considerably. So, why has only this period been considered? And, is it possible that cross-country comparison would have influenced the results?

    Moreover, the size of the sample analyzed is not specified, nor is the sector(s) to which the companies belong (which is extremely important considering that the research analyses changes in productivity). This lack is also found in the analysis for blockchain development.

  • Finally, in my opinion it is important that the authors expound and describe the nature of the implications of this research.

  • Thus, until integrations are made, the paper can not be published in this form.

  •  
  •  

Round 2

Reviewer 3 Report

Ok.

take in consideration the impact of blockchain in subsequent years, even considering other variables, such as economic and fiscal variables
